# Theory informed, experiment based, constraint on the rate of autoxidation chemistry – An analytical approach

Lukas Pichelstorfer[1,2], Simon P. O'Meara[3,4], Gordon McFiggans[3]

[1]pi-numerics, Neumarkt am W., 5202, Austria
[2]Institute for Atmospheric and Earth System Research/Physics, University of Helsinki, 00560 Helsinki, Finland
[3]Department of Earth and Environmental Sciences, University of Manchester, Manchester, M13 9PL, UK
[4]National Centre for Atmospheric Science, UK

*Correspondence to*: Lukas Pichelstorfer (office@pi-numerics.com)

**Abstract.** Autoxidation is a key process that transforms volatile organic compounds into condensable species, thereby significantly contributing to the formation and growth of airborne particles. Given the enormous complexity of this chemistry, explicit reaction mechanisms describing autoxidation of the multitude of atmospherically relevant precursors may appear out of reach.

The present work suggests an alternative solution path: based on theoretically suggested key reaction types and the recent advances in mass spectroscopy, an analytically-based approach to constrain lumped autoxidation reaction schemes is presented. Here, the method is used to equip an autoxidation reaction scheme for α-pinene with rate coefficients based on the interpretation of simulated data. Results show the capability to recover the rate coefficients with a maximum error of less than 1% for all reaction types. The process is automated and capable of determining roughly $10^3$ rate coefficients per second when run on a PC. Currently, the method is applicable to chemical systems in a steady state, which can be established in flow reactors. However, extending the concept to allow analysis of evolving systems is part of ongoing work.

## 1 Introduction

Volatile organic compounds (VOC) have been detected in every environment around the globe (Blake and Blake, 2003). In the atmosphere, they react with radicals or sunlight to form oxidised species fuelling the formation of secondary organic aerosol (SOA; Hallquist et al., 2009). SOA is considered to substantially contribute to the ambient aerosol particle mass and number (Jimenez et al., 2009), thereby impacting air quality (Daellenbach et al., 2020) and climate (Calvin et al., 2023).

Descriptions of VOC oxidation typically consider the decomposition of the carbon structure, ultimately leading to the formation of $CO_2$ (Kroll and Seinfeld, 2008). However, in the last decade, the process of autoxidation, leading to accumulation of oxygen at the carbon centered radicals, was found relevant under atmospheric conditions (Jokinen et al., 2014; Rissanen et al., 2014). Prior to this, autoignition (i.e., gas-phase autoxidation) was associated with combustion processes (Cox and Cole, 1985; Wang and Sarathy, 2016). This process can be described as follows: alkoxy (RO•) and

peroxy radicals ($RO_2\bullet$) undergo sequential intra-molecular hydrogen abstraction and addition of $O_2$ to form a $RO_2\bullet$ species (Crounse et al., 2013; Mentel et al., 2015; Vereecken and Peeters, 2010). The advancing enrichment of oxygen alters the species properties: the molecules polarity and weight increases, leading to a decrease of the saturation vapor pressure.

The present manuscript introduces a novel, semi-empirical way to quantify autoxidation reactions. Autoxidation under ambient conditions currently lacks a VOC-specific mechanistic description and consequently, is not well represented, even in detailed atmospheric chemistry models, such as the Master Chemical Mechanism (MCM v.3.3.1: Jenkin et al., 2003; Saunders et al., 2003) or Generator for Explicit Chemistry and Kinetics of Organics in the Atmosphere (GECKO-A: Aumont et al., 2005; Camredon et al., 2007). As a result, even detailed aerosol-chemistry models applying the MCM additionally require a general description of autoxidation chemistry (Clusius et al., 2022; O'Meara et al., 2021; Roldin et al., 2014). Large scale models, typically, may consider the contribution of autoxidation implicitly by applying empirical mass (or low volatility species) yields upon the oxidation of a VOC species (Bergman et al., 2022). These yields are obtained in experimental setups and represent the hitherto incomprehensible complexity of the autoxidation chemistry by a single number or function: the aerosol particle mass or yield of low volatility species that forms upon the consumption of a certain amount of VOC by a specific oxidant (e.g.: (Kroll et al., 2005; Kroll and Seinfeld, 2008). It is unclear whether and how such experimentally-derived particle mass yields can represent process dependencies that may occur under atmospheric conditions.

Mechanistic approaches, aiming to understand the rearrangements of the specific molecular structures leading to the oxidative molecular growth are limited by the enormous number of chemical species formed in the chemistry: the number of molecular structures present during the degradation of n-alkanes is roughly $10^{nC-1}$ (with nC being the number of carbon atoms in the parent structure; Aumont et al., 2005). This omits the additional complexity contributed by autoxidation (Bianchi et al., 2019; Franzon et al., 2024; Rissanen et al., 2015). To explicitly predict the evolution of the chemistry, the most important reaction pathways would need to be identified and quantified. Owing to the sheer number of species such an approach is not possible.

Clearly, any approach aiming to capture this overwhelmingly complex system, requires simplifications. Such a simplification is provided by the volatility basis set-approach (VBS; Donahue et al., 2006, 2011; Schervish and Donahue, 2020) that distributes the products of simplified (autoxidation) chemistry into volatility bins. The formal scale (i.e., relatively few equations) of the method allows for application in large-scale models (e.g., Shrivastava et al., 2024; Tsimpidi et al., 2010). However, due to the lumping of the chemistry, it may be difficult to relate the model-parameters to actual, measurable quantities. Nevertheless, the VBS approach enables consideration of autoxidation-specific aspects in atmospheric models. The method is highly valuable in that it will enable any future, mechanistic model of the (autoxidation) chemistry to be reduced to a similar form. However, the required mechanistic underpinning to do so, is, as yet, missing and a promising methodology to overcome remains to be proposed.

A recent study (Roldin et al., 2019) considers a collective behaviour of the peroxy radicals of similar atomic composition. This greatly reduces the complexity of the system whilst still allowing description of key autoxidation chemistry (autoxidation of RO• and RO$_2$•) and competing reactions (mainly, RO$_2$• + HO$_2$•/NO•/RO$_2$•, including adduct formation).

This approach requires inputs for mechanistic constraint; i.e., it needs to be related to some observations in order to derive rate coefficients for the lumped reactions. The input data requires sampling at high atomic resolution, with known (ideally uniform) instrument sensitivity and molecular specificity.

It can be assumed that the nitrate chemical ionization mass spectrometer (NO$_3^-$ CIMS) is able to provide such data (Ehn et al., 2014; Jokinen et al., 2012):

1) it is considered highly sensitive to the most oxidised species (Hyttinen et al., 2018)

2) it provides information on the chemical composition of the detected species

3) it is an online tool providing a timely resolution enabling to capture the evolution of closed shell species and peroxy radicals

Authors, previously, have interpreted the signal intensity of the NO$_3^-$ CIMS as concentration enabling determination of rate coefficients for accretion product formation from RO$_2$• + RO$_2$• reactions (Berndt et al., 2018a, b) or determination of ELVOC yield (Ehn et al., 2014). In the present work we, too, use the interpretation of the NO$_3^-$ CIMS signal as concentration. We systematically employ the approach by Roldin et al. to set up chemical reaction schemes describing autoxidation chemistry using a recently developed tool (Pichelstorfer et al., 2024).

A new, analytically based, method to determine the rate coefficients for autoxidation reaction schemes is presented. We mathematically demonstrate that the new method can be used to find rate coefficients that can plausibly describe the chemical species' evolution based on simulated mass-spectra. The method compares concentrations of detected species (at atomic resolution, e.g., by NO$_3^-$ CIMS) and their change rates (with respect to time) to isomers that exist in a proposed, lumped reaction scheme (Pichelstorfer et al., 2024). Assumptions on the relative magnitude of the isomer-forming rate coefficients and a description of physical change parameters (e.g., dilution, wall interaction) allow calculations of reaction rate coefficients of the proposed scheme.

## 2 Methods

In order to relate the evolution of the CIMS signal (equivalent to measured concentration) to rate coefficients of an autoxidation scheme, a list of reactions likely governing the chemistry must be assumed. This list includes the autoxidation reaction of alkoxy and peroxy radicals, as well as competing reaction pathways leading to the formation of closed shell (CS) and RO• species.

Generally, each peroxy radical can interact with any other peroxy radical to either form accretion products (ROOR': Berndt et al., 2018b), alkoxy radicals (Crounse et al., 2013), or form hydroxyl (ROH) and carbonyl (RC=O) species in a complex chemical reaction (see: Orlando and Tyndall, 2012; Salo et al., 2022). In the present work, the accretion product formation is considered explicitly, as the product contains information on the reactants (i.e., the atomic composition of ROOR') necessary for mechanistic traceability:

$$RO_2• + RO'_2• \quad \rightarrow \quad ROOR' (+ O_2) \tag{R1}$$

However, the formation of RO•, ROH and RC=O only provides information on one reactant (since the product doesn't contain information on the second reaction partner R'). Accordingly, they are described in a lumped way, as the reaction with the sum of peroxy radicals in the system (i.e., $\Sigma[RO_2•] = [RO_{2,1}•] + [RO_{2,2}•] + [RO_{2,3}•] + ... + [RO_{2,n}•]$ with n different peroxy radicals in the system):

$$RO_2• + \Sigma RO_2• \quad \rightarrow \quad ROH \quad (+ O_2 + \Sigma RO_2•) \tag{R2a}$$
$$\rightarrow \quad RC=O (+ O_2 + \Sigma RO_2•) \tag{R2b}$$
$$\rightarrow \quad RO• \quad (+ O_2 + \Sigma RO_2•) \tag{R2c}$$
$$\rightarrow \quad \text{fragmentation products} \quad (+ \Sigma RO_2•) \tag{R2d}$$

Reactions R2a to R2d do change the reacting peroxy radical's concentration but have only minor effect on the $\Sigma RO_2•$. This approach does not conserve the number of H atoms in the system but greatly reduces the number of those reaction equations difficult to constrain. Details on this approach can be found elsewhere (Jenkin et al., 2003; Saunders et al., 2003).

Other reaction pathways of the $RO_2•$ include the interaction with $HO_2•$ (Orlando and Tyndall, 2012). In the present work we consider the following product pathways:

$$RO_2• + HO_2• \quad \rightarrow \quad ROOH \tag{R3a}$$
$$\rightarrow \quad RO• + O_2 + OH• \tag{R3b}$$
$$\rightarrow \quad \text{fragmentation products} \tag{R3c}$$

Further, the interaction of $RO_2•$ and NO• (Orlando and Tyndall, 2012) is considered:

$$RO_2• + NO• \quad \rightarrow \quad RONO_2 \tag{R4a}$$
$$\rightarrow \quad RC=O + OH• + NO_2 \tag{R4b}$$
$$\rightarrow \quad RO• + NO_2 \tag{R4c}$$
$$\rightarrow \quad \text{fragmentation products} \tag{R4d}$$

Equations R1 to R4, mainly represent reactions which can terminate autoxidation, though there are some branches forming RO• which allow autoxidation to continue (R2c, R3b, R4c). In the present work, the intra-molecular abstraction of hydrogen and subsequent addition of $O_2$ comprising the autoxidation channel (Bianchi et al., 2019) is represented by R5a:

$$RO_2\bullet \quad \text{H-shift} \quad \rightarrow \quad RO_2'\bullet \quad\quad\quad\quad \text{(R5a)}$$

In cases where the abstracted H is in alpha-position to a hydro(pero)xy functional group, a carbonyl group is formed to terminate the autoxidation (R5b; Orlando and Tyndall, 2012):

$$RO_2\bullet \quad \text{H-shift} \quad \rightarrow \quad RC=O + HO_2\bullet \quad\quad\quad\quad \text{(R5b)}$$

Any molecular rearrangements leading to the fragmentation of the carbon chain are represented by reaction R5c:

$$RO_2\bullet \quad \text{H-shift} \quad \rightarrow \quad \text{fragmentation products} \quad\quad\quad\quad \text{(R5c)}$$

Due to their relatively short lifetime (at most, 1 ms in the lower atmosphere; Orlando et al., 2003) and resulting low concentrations, bimolecular reactions (except for $O_2$ addition upon autoxidation) of alkoxy radicals is neglected in the current approach. Analogous to $RO_2\bullet$, RO• can undergo autoxidation along with the same competing channels (Bianchi et al., 2019; Vereecken and Peeters, 2010):

$$RO\bullet \quad \text{H-shift} \quad \rightarrow \quad RO_2\bullet \quad\quad\quad\quad \text{(R6a)}$$
$$\rightarrow \quad RC=O \quad\quad\quad\quad \text{(R6b)}$$
$$\rightarrow \quad \text{fragmentation products} \quad\quad\quad\quad \text{(R6c)}$$

Note that the rearrangement of the alkoxy radical (roughly $10^3$ to $10^7$ s−1) is considered to happen much faster, compared to the peroxy radical (mainly below subsecond with few exceptions) (Crounse et al., 2013; Praske et al., 2018; Vereecken and Nozière, 2020; Vereecken and Peeters, 2010).

The concentrations of the (unfragmented) carbon centred radicals and their closed shell reaction products are obtained from $NO_3^-$ CIMS measurements. However, there are other species in the reactions R1 to R6 that require different determination methods. In the present work, the concentrations of $HO_2\bullet$, NO•, and the $\Sigma RO_2\bullet$ (including all peroxy radicals of the chemical system) are determined by modelling. The MCM, which describes the tropospheric degradation of a variety of hydrocarbons, is deployed for this purpose. Concentrations of fragmentation products, due to the unknown fragmentation

pathways, are estimated (RO reaction branch: lower limit is 0, upper limit is determined by the kinetic limitation for the considered reaction, e.g. $RO_2+HO_2$; for R6c, the upper limit is the production of RO species).

To greatly simplify the solution of the mathematical description of the autoxidation chemistry, we consider a well-mixed, steady-state system (i.e., there is no timely variation in the $NO_3^-$ CIMS signals). As a result, the differential equations governing the evolution of the chemical species reduce to differences that can be represented by linear equations:

$$\frac{dC}{dt} \approx \sum^i \frac{\Delta C_i}{\Delta t} = 0 \qquad \text{equ. (1)}$$

where i is the number of processes impacting the concentration (C) of a chemical species. A similar experimental setup was investigated by Roldin et al. (2019) to derive rate coefficients for an autoxidation chemistry scheme. The procedure of rate fitting to reproduce the mass peaks detected by CIMS represents a considerable workload (several weeks to months for a

165 single VOC system). In the present work, we approach the problem differently to enable automation: instead of incrementally optimizing rate coefficients to meet a mass-peak distribution at a certain point in time, we set up the linear set of equations describing the steady state. Each solution of the set of linear equations represents a valid mathematical solution to the problem (i.e., a set of reaction rate coefficients explaining the CIMS measurement). The recently published automated alkoxy-/peroxy-radical autoxidation mechanism framework (autoAPRAM-fw; Pichelstorfer et al., 2024) is capable of

170 creating autoxidation reaction schemes in an automated fashion for any VOC system. Due to the large number of equations involved and the potentially numerous solutions, we apply the autoAPRAM-fw to create a description of the autoxidation chemistry (without assigning rate coefficients): the "APRAM scheme". Briefly, the framework uses input peroxy radical names, their atomic composition and the information on the reaction types considered to create a list of reactions including product names (with likely atomic composition) and default reaction rate coefficients. For details on the formalism, we refer

to the original article.

Consideration of the physical boundary conditions of the experimental configurations (wall-interaction, influx, outflux) allows the linear set of equations relating changes in the detected mass peak to the chemical change terms (obtained from the reaction scheme) to be constructed. The unknowns (k) represent the missing rate coefficients:

$$\frac{\Delta C(CHON)_x}{\Delta t} = \sum_{CHON_i = CHON_x} \frac{\Delta C_i}{\Delta t} = \sum_{m,n \to i} k_m * \left[ R_{1,m} \right] * \left[ R_{2,n} \right] + \sum_{u \to i} k_u * \left[ R_u \right] - \sum_i \frac{\Delta C_{i,phys}}{\Delta t} \qquad \text{equ. (2)}$$

Where $\Delta C(CHON)_x$ is the change in concentration in $cm^{-3}$ of CIMS-detected species with atomic composition CHON. For each experimentally determined atomic composition, a list of model isomers i and related concentration changes $\Delta C_i$ can be assigned. This list may be empty in case the chemical scheme does not cover the experimentally determined species. The

185 first term on the right-hand side of eq. 1 represents all changes to the related chemical scheme product isomers'

concentration (rate coefficient k, reactants $R_1$, $R_2$) due to bimolecular reactions. The second term represents unimolecular reactions and the third term summarizes any physical changes to the scheme products' concentrations $C_i$ (i.e., in the present work: a dilution flow). A detailed description of how the equation (2) is set up and solved for each species of the observed spectrum can be found in the supporting information.

## 2.1 Key aspects of the approach

**$RO_2\bullet$.** Clearly, the peroxy radical is a key species in the description of the autoxidation chemistry (Ehn et al., 2014; Goldman et al., 2021). As a result, determination of the $RO_2\bullet$ abundance is an important task in constraining the chemistry. In the present work we use simulated data mimicking the $NO_3^-$ CIMS experimental data to derive the $RO_2\bullet$ concentrations. In experimental applications, it is assumed that all highly oxygenated $RO_2\bullet$ species are quantified with equal sensitivity (Hyttinen et al., 2018).

**$RO\bullet$.** Alkoxy radicals, owing to their short life-time (roughly 0.1 ms in the earth's atmosphere (Orlando et al., 2003)), are not measured and, accordingly, are not a target of the reaction rate analysis. Instead, the fraction of reaction products branching towards the alkoxy is estimated (reactions: 2c, 3b and 4c). Similarly, the reaction product branching of reaction 6 is estimated in order to investigate the potential effects of the alkoxy pathways on the $RO_2\bullet$ abundance and closed shell species formation. The $RO\bullet$ chemistry a) allows the $RO_2\bullet$ - oxygen amount to switch between odd to even numbers and b) enables the autoxidation to proceed effectively at low $RO_2\bullet$ concentrations via reactions R2c,R3b or R4c followed by R6a.

**Closed shell species (CS).** Setting up the (chemical) formation and (physical) loss balance for closed shell species allows for the determination of rate coefficients applied in reaction types (1), (2a-b), (3a), (4a-b), (5b) and (6b).

**Fragmentation products.** Although direct inclusion of reactant-specific fragmentation products is beyond the scope of this work, the effect of this reaction can be investigated. For mass balance, the larger the estimated flux through fragmentation (see reaction equations 2d, 3c, 4d, 5c and 6c), the higher the production of the reagents via reaction 5a and 6a must be. Accordingly, the fragmentation branching has a potentially large impact on the H-shift rates determined with the suggested method.

**H-shift rates**. Deriving hydrogen-shift rates requires: determination of the CS-forming rate coefficients, estimation of the loss through fragmentation reactions and calculation of potential contributions via the alkoxy pathways. $RO_2\bullet$ H-shift rates leading to the formation of another $RO_2\bullet$ species with an oxygen atom number raised by 2 (i.e., reaction type 5a - autoxidation) can be calculated from a balance equation (see also Fig. 1):

$$\frac{\Delta C\left(RO_{2,k}\bullet\right)}{\Delta t}=k_{H-shift}C\left(RO_{2,i}\bullet\right)+S_k-L_k-k_{H-shift}C\left(RO_{2,k}\bullet\right) \qquad \text{equ. (3)}$$

Where $C(RO_{2,k}\bullet)$ is the concentration of the peroxy radical k. $S_k$ is a source term considering all potential contributions including wall sources as well as autoAPRAM-RO• autoxidation, MCM-RO• and -RO$_2$• pathways. $L_k$ is a loss term (physical as well as chemical but not including RO$_2$• autoxidation). The $RO_{2,k}\bullet$ is the product of the $RO_{2,i}\bullet$ undergoing H-shift and O$_2$ addition.

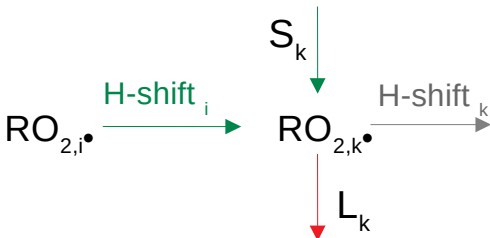

**Figure 1: Balance of the in- and out-flux of the $RO_{2,k}\bullet$ to derive the H-shift rate for $RO_{2,i}\bullet$.**

The determination of H-shift reaction rate coefficients of a $RO_{2,i}\bullet$ forming $RO_{2,k}\bullet$ requires a) the knowledge of closed shell and fragmentation species formation rate coefficients, as well as b) the H-shift rate of the $RO_{2,k}\bullet$. Accordingly, equation (2) is solved starting with the closed shell species of largest molecular mass (more information on this can be found in the SI - "On sorting the input CIMS data"). Next, fragmentation and alkoxy pathways are considered enabling the H-shift rate to be assigned.

### 2.2 The effect of RO$_2$• species not part of the APRAM scheme

In the present work, the autoxidation chemistry scheme is created by the autoAPRAM-fw. It does not include the degradation of the VOC upon oxidation, except for a few reactions resulting in fragmentation of the reactant (3c, 4d, 5c and 6c). If autoxidation products from the autoAPRAM-fw were to comprise a small fraction of the total abundance of RO$_2$•, reactions with the ΣRO$_2$• may be dominated by species outside the autoxidation scheme, depending upon their reaction rates. This potential dominance is neither guaranteed nor obvious, as the respective reaction rate coefficients may be one or two orders of magnitude different for these RO$_2$• formed by the fragmentation of the carbon chain or by oxidation of a different parent VOC (Jenkin et al., 2019). Knowledge of the abundance of non-APRAM RO$_2$• is required to equip the ΣRO$_2$• reactions with rate coefficients. Further, the lowest oxygen-number containing species considered by the autoAPRAM-fw may not be formed within the framework. Accordingly, the present approach requires a model describing the degradation of the VOC (i.e., oxidation reactions leading to fragmentation of the VOC carbon chain, ultimately forming CO$_2$). This process is characterised by a sequence of oxidation reactions forming radicals (R•, RO• and RO$_2$•) that decompose or form closed-shell species. Some of those radicals may undergo autoxidation, thereby forming species considered by the APRAM

chemical scheme. In the present work, MCM serves for degrading the VOC, creating radicals and estimating concentrations of $RO_2\bullet$ (i.e., species that are not considered in the APRAM scheme). Here, MCM forms the enabling assumption to solve the chemical equations.

To estimate the flux from a specific (MCM) species into the APRAM scheme, the consumption of the product peroxy radical $RO_{2,k}\bullet$ (by dilution, wall loss, and chemical reactions) is balanced with potential formation terms (see Fig. 1). In cases where H-shift of $RO\bullet$ and $RO_{2,i}\bullet$ (both part of the autoxidation scheme) may not compensate for the consumption of the $RO_{2,k}$, additional production terms are required. Particularly, at the lower molecular mass-end of the mass spectrum (where no $RO_{2,k}$ formation reactions via the autoAPRAM scheme exist), the consumption of a radical has to be entirely balanced by an influx. Determination of the required influx is made possible by the order of calculation: as presented in the methodology, the scheme is equipped with rate coefficients starting at the upper observed product mass as is explained in the next section ("Solving the equations"). In the final step, a required influx may be calculated as described below, with units of molecules $cm^{-3}s^{-1}$.

In the present autoxidation chemistry, the production flux of $RO_{2,k}\bullet$ can originate from a) $RO\bullet$ and $RO_2\bullet$ undergoing autoxidation as well as b) production of the species by reactions not covered by the autoxidation chemistry. The initiation flux probability (P) via the $RO\bullet$ and $RO_{2,i}\bullet$ pathways to balance the loss of $RO_{2,k}\bullet$ is considered:

$$P(RO\bullet) = 1 - (L_{chem} + L_{phys} - S_{in}) / (L_{chem} + L_{phys} + S_{in}) \qquad \text{equ. (4)}$$

$$P(RO_{2,i}\bullet) = 1 - C(RO_{2,i}\bullet) * k_{autox., high} / (L_{chem} + L_{phys} - S_{in}) \qquad \text{equ. (5)}$$

Where L are chemical ("chem") and physical ("phys") loss terms of the $RO_{2,k}\bullet$ (see Fig. 1 for relation $RO_{2,i}\bullet$ and $RO_{2,k}\bullet$). $S_{in}$ is a source term of the $RO_{2,k}\bullet$ via the alkoxy pathway. $C(RO_{2,i})$ is the concentration of the $RO_{2,i}\bullet$ and $k_{autox,high}$ is a (structure independent) estimated upper limit for the autoxidation rate of $RO_{2,i}\bullet$. In the current work we assumed $k_{autox,high} = 2$ $s^{-1}$. We chose this conservative upper limit as most H-shifts are significantly slower, although considerably higher H-shift rates have been reported in recent literature (Iyer et al., 2023). The initiation flux probabilities of $RO\bullet$ and $RO_2\bullet$ relate to equ. (3) as all are based on considerations of mass conservation. While equ. (3) is explicitly balancing mass fluxes, P is meant to highlight the imbalances in fluxes by considering a) mass fluxes via the RO pathway for $P(RO\bullet)$, and b) expected limitations of the fluxes from $RO_2\bullet$ within the autoAPRAM scheme for $P(RO_2\bullet)$.

Where probabilities $P(RO\bullet)$ and $P(RO_2\bullet)$ both approach 1, an initiation flux to compensate the loss of $RO_{2,k}\bullet$ very likely is required. The magnitude of the influx can be determined as the difference between the loss of $RO_{2,k}\bullet$ and the formation via $RO\bullet$ and $RO_{2,i}\bullet$ paths.

Where the probabilities $P(RO\bullet)$ and $P(RO_2\bullet)$ both are << 1, influx is not prerequisite but still possible. $P(RO\bullet)$ and $P(RO_2\bullet)$ are defined in the range [0;1]. negative results indicate a likely initiation flux for $P(RO\bullet)$ and an unlikely initiation for

P($RO_2\bullet$). In this case, H-shift of RO• and $RO_2\bullet$, together with the initiation-flux form an equation with three unknowns. Accordingly, the potential solution space is a 2-D surface. Calculated influx probabilities for all $RO_2\bullet$ species considered can be found in the SI (see table S2 in the supporting information).

### 2.3 Solving the equations

Rate coefficients are assigned by applying the newly created, semi-automated "autoCONSTRAINT" tool (see Fig. 2). The autoCONSTRAINT tool reads in a chemical scheme with no rate coefficients, together with a mass spectrum obtained e.g. with $NO_3^-$ CIMS. The $\Sigma RO_2\bullet$, $HO_2\bullet$, NO• concentrations and boundary conditions (temperature, wall loss and dilution) are additionally required as input parameters.

Based on the autoAPRAM scheme, autoCONSTRAINT derives the linear set of equations (i.e., equ (2)), describing the

chemical and physical change terms for the $NO_3^-$ CIMS-detected peak-list. Note that not all peaks detected with CIMS may be covered by the scheme. Analogously, not all of the species from the APRAM scheme may be detected with the CIMS. In both cases, a rate coefficient related to these species is not accessible.

Typically, each CIMS-detected peak relates to several change terms (production and loss reaction pathways). Consequently, there are several unknowns, as every chemical change term of equation (2) features a rate coefficient. To enable solution for

the rate coefficients, the number of unknowns is reduced by providing a guess on the relative magnitude of the rate coefficients. For example, in case of three unknown rate coefficients, an estimate for two of them in relation to the third must be provided. As a result, the solutions are typically not unique. However, where they are chemically meaningful (i.e., the rate coefficients are in the interval starting from zero to the kinetic/theoretic limit), they represent a pool of equally valid solutions. Provision of relative values for rate coefficients can account for both experimental and theoretical findings.

A detailed description on the construction and solution of equation (2) for each species of the observed spectrum can be found in the supporting information (note that the mass spectrum - input may also be of simulated nature and does not necessarily have to be experimental).

Equation (3) depicts the dependence of the H-shift rate of a species i on the loss terms of $RO_{2,k}\bullet$ (which is increased in mass by 2 oxygen atoms). Consequently, determination of the rate coefficients starts at the upper end of the mass spectrum. This

enables the chemical consumption of every $RO_2\bullet$ species to be determined before the autoxidation rates are calculated. A detailed depiction of these considerations can be found in the SI (section "On sorting the input CIMS data").

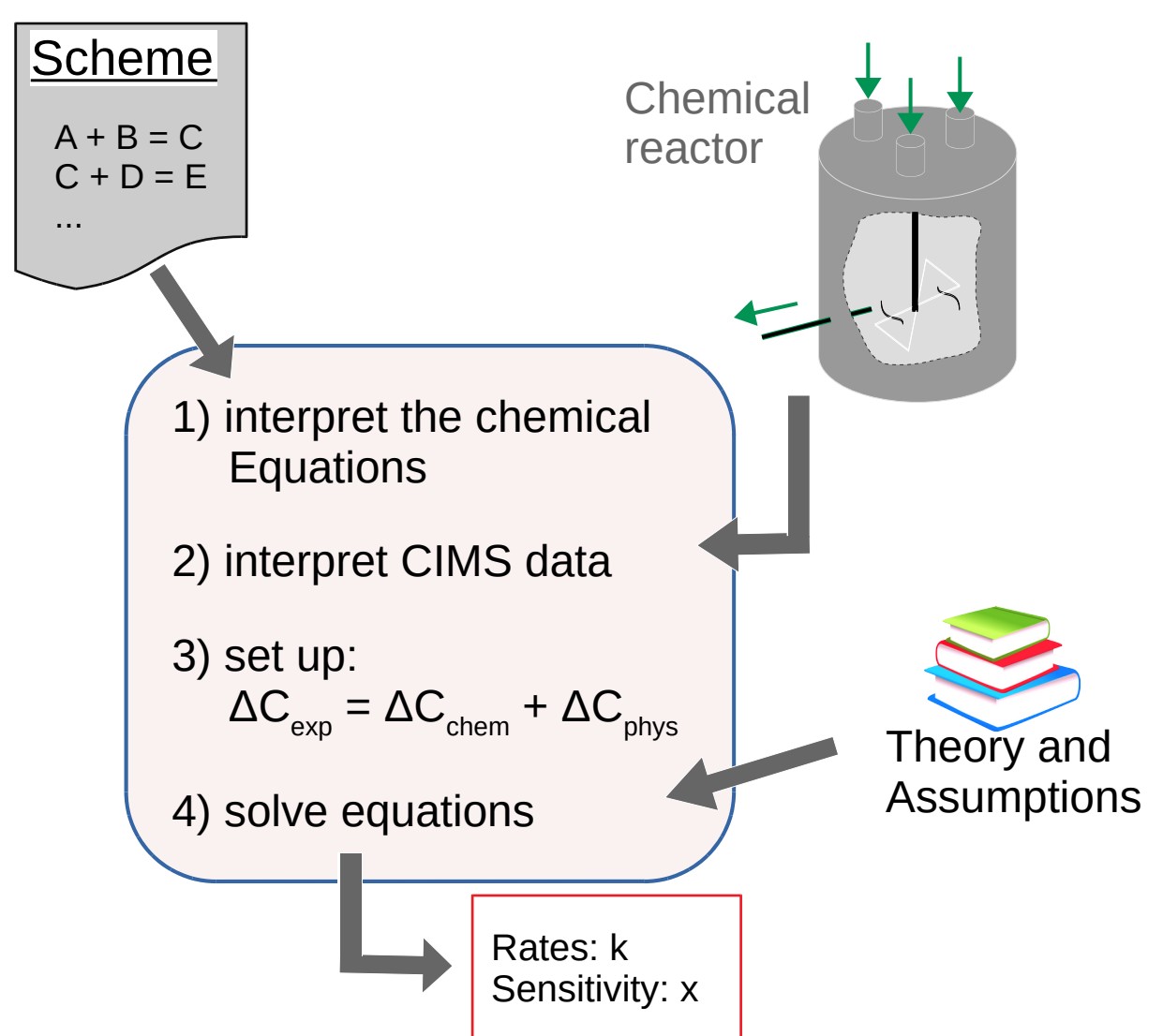

**Scheme**

A + B = C
C + D = E
...

Chemical reactor

1) interpret the chemical Equations

2) interpret CIMS data

3) set up:
$\Delta C_{exp} = \Delta C_{chem} + \Delta C_{phys}$

4) solve equations

Theory and Assumptions

Rates: k
Sensitivity: x

*Figure 2: the autoCONSTRAINT workflow. The autoCONSTRAINT module reads in chemical schemes and experimental results (including high resolution CIMS data). The, typically, underconstrained equations solved by adding theoretic knowledge or*
305 *assumptions. The output covers rate coefficients k and an estimate on their sensitivity x.*

We examined modelled data to showcase the capability of autoCONSTRAINT's to reproduce rate coefficients from an input chemistry scheme. For this demonstration, a scheme mimicking the autoxidation of α-pinene was created (i.e., the αpin-test scheme) applying the autoAPRAM-fw. The full scheme can be found in the supporting data (see supporting data: PyCHAM/input/apinene_autoPRAM_V01_scheme.dat). Roldin et al. (2019) provided a possible solution for the same
chemical system in a similar autoxidation scheme. However, the species covered in the present approach do not fully match

those covered by Roldin et al. (2019). The difference is the description of the adduct formation: in the work of Roldin et al., the adduct formation is lumped, while it is specific to reagent products in the αpin-test scheme. As a result, the scheme presented in this work should be considered fully independent from Roldin's scheme and its sole purpose is to create data to be analysed using autoCONSTRAINT. The reason to choose α-pinene is to illustrate the model's potential to deal with systems of similar complexity.

We followed the steps described in Fig. 2 to apply the method:

1) the αpin-test scheme (without rate coefficients) describing α-pinene autoxidation was input to autoCONSTRAINT and the reactions are interpreted. Optionally, the model reports the read-in reactions and how they are interpreted (see file "chem_interpretation.txt" in the SI).

2) To mimic "experimental data", PyCHAM model (O'Meara et al., 2021) simulations were conducted in flow reactor mode (i.e., there is in- and outflux to the chamber, a characteristic residence time and losses by dilution). Details of the simulation setup can be found in the SI (section "PyCHAM inputs"). The point in time for analysing the change in CIMS signal was chosen 6 hours after the start of the simulation. At this time, the chemical system can be approximated as being in a steady state. The PyCHAM model results, which include full information on the isomeric distribution were converted to high resolution CIMS data format (i.e., the data includes information on the atomic composition only and therefore isomers can not be distinguished) and provided as an input to autoCONSTRAINT. Additional (simulated) data used in the analysis describe the concentrations (and change rates) of species not part of the αpin-test scheme (see section 2.2 for more information).

After sorting the CIMS peaks in descending mass order, species from the αpin-test scheme were assigned to their respective mass-spectrum peaks based on their atomic composition. This allows assembly of the chemical and physical change terms for each peak in the mass spectrum (examples can be found in the SI – section "Exemplary solution of equation 2").

3) MCM simulations of α-pinene oxidation were conducted to provide an estimate of the peroxy radical isomeric distribution of species not covered by the αpin-test scheme in addition to estimates of the concentrations for $NO\bullet$, $HO_2\bullet$ and the $\Sigma RO_2\bullet$ (since these quantities were not measured in this example).

4) The equations were solved after providing an estimate of the relative magnitude of the individual reaction types' rate coefficients. The relative values applied were provided as an input.

## 3 Results & Discussion

### 3.1 Reproduction of k-rates from known scheme

The reproduction of rate coefficients from the αpin-test scheme are summarized in Fig. 3 as well as in SI Figs. S4-S7. The deviation from the αpin-test scheme rate coefficients is below 0.2 % for all rate coefficients except for accretion products which show a maximum error of 0.57% and a mean absolute error of 0.05%.

Based on the choice of relative magnitude of the individual rate coefficients, the contribution of a single reaction to the sum of chemical changes can be determined. An example is given in the SI (see section "Setting up equ (2) for mass peak 214:"). As the contribution reduces, the dependence on the choice of relative rate coefficient increases. For example, reaction channel A contributes to 99% of the chemical changes of an observed product and reaction channel B contributes 1%. When doubling the contribution of reaction B, its rate roughly doubles but the rate of A remains almost the same (changing by roughly 1%). Accordingly, the colour code in Fig 3 can be interpreted as a sensitivity parameter.

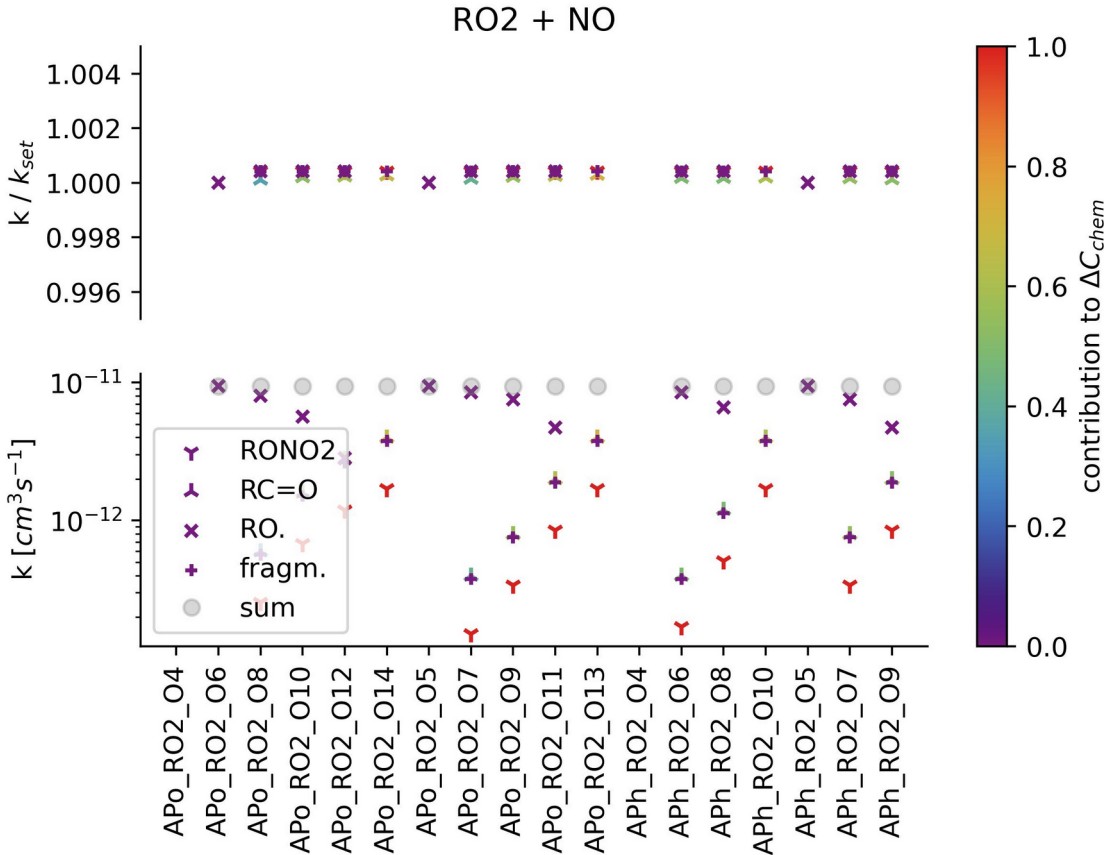

**Figure 3:** *$RO_2$ radicals (names are given by the abscissa) react at a rate (denoted on the lower ordinate) to form RONO2 (downward facing triangle), alkyl species (upward facing triangles), fragmentation species (plus) and alkoxy radicals (crosses). The sum of the individual reaction channel's rates is shown by a circle. The precision of rate coefficient recovery from the (simulated) mass spectrum is shown on the upper ordinate. The colour code denotes the calculated contribution of the reaction to the change (by means of chemical reactions) in CIMS mass peak concentration.*

Figure 4 presents an example of the impact of the estimate of relative rate coefficients on an estimated rate coefficient (in this example the rate coefficient of $RO_2$•(Apo_RO2_O10) + $HO_2$• →ROOH). The parameter of interest (see abscissa) is varied by -50%, -25%, +25%, +50 as indicated by the colour bar. Variation of guess for (R3a) (i.e., $RO_2$• + $HO_2$• → ROOH)

results in a change of the rate coefficient by -23% to +11%. The guess on the reaction (R4) (RO$_2$• + NO•) does not affect the product ROOH. For the variation of reaction (R2) (RO$_2$• + ΣRO$_2$•), an indirect effect on (R3a) is observed. Reduction of the guess-value by 50% results in an increase of k$_{ROOH}$ by roughly 18%, while increasing the guess for (R2) by 50% reduces the k$_{ROOH}$ by 17%. The wall loss has a direct effect on the rate coefficient k$_{ROOH}$: changing the estimate on the wall loss by a given amount results in a similar change of the rate coefficient. Assuming that the conversion from CIMS signal to concentration can be approached by a constant value (i.e., uniform sensitivity of the instrument) for all molecules of the autoAPRAM reaction scheme, there is no effect of this constant value on the rate coefficient k$_{ROOH}$. However, the conversion constant affects the reactions that involve RO$_2$•-RO$_2$'• reactions forming adducts (R1), carbonyls (R2b) or alcohols (R2a) due to the square dependence on the RO$_2$• concentrations: in other reactions (bi- or unimolecular) considered in this work, a single reagent (the RO$_2$•) and the product are sampled with a certain sensitivity. Consequently, this sensitivity, if uniform, does not matter as long it is high enough to detect the peaks properly. The potential reaction partners (NO • and HO$_2$•) are not subject to this sensitivity as they are determined by other means. However, where both reagents (RO$_2$• and RO$_2$'•/ΣRO$_2$•) are subject to this sensitivity, the product forming rate coefficient is linearly dependent on it.

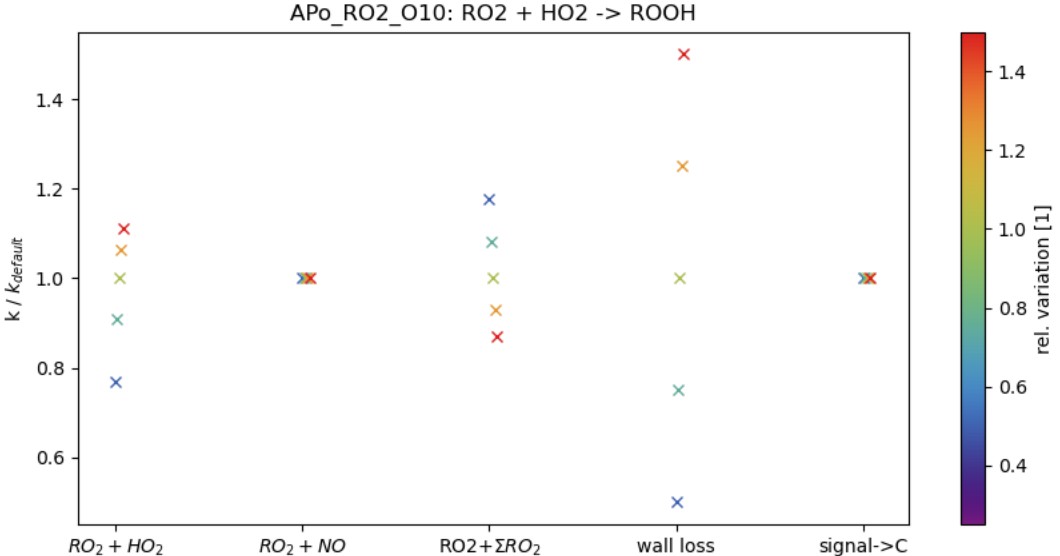

**Figure 4: effect of variation of the rate-coefficient guess (RO$_2$•+HO$_2$•; RO$_2$•+NO•; RO$_2$•+ΣRO$_2$•), the "wall loss" and the signal to concentration conversion ("signal → C") on the formation of ROOH species via reaction (R3a). The colour bar indicates the magnitude of variation of the guess.**

### 3.2 Interpretation of the result

All results determined with the proposed method, mathematically, represent a valid solution (i.e., a full set of rate coefficients for the scheme) to the problem. Several of the solutions found may satisfy the boundary conditions (e.g., the rate

coefficient may not exceed the kinetic limit). These are all mechanistically plausible and there is no additional constraint available to determine which of the solutions is the best. To discriminate between them, additional information is required. As the mathematical nature of the drafted approach is a linear set of equations, additional information can be gained by considering multiple sets of CIMS data determined under different chemical regimes.

### 3.3 On determining multi-experiment based, realistic rate coefficients

The confidence in the calculated rate coefficients is dependent on the corresponding reaction's dominance among the chemical terms in equation 2. For example, we can determine with confidence the rate of reaction $RO_2\bullet + HO_2\bullet$ when the product, ROOH, comprises a large share of the product isomers. As discussed in section 3.1 "Reproduction of k-rates from known scheme", the value of the rate coefficient is only weakly dependent on the choice of relative k if the reaction channel is dominant. As a result, data from an experiment covering just one point in the chemical space provides relatively poor

constraint over rate coefficients for minor reactions at that point. Consequently, data sets gathered under a range of chemical regimes (high $HO_2\bullet$, high $NO_x$, high $RO_2\bullet$ – low $NO_x$) can be jointly analysed to more effectively quantify all rate coefficients.

The chemistry under investigation is clearly complex and the proposed approach will never be likely to cover all reaction channels and species. However, we show that plausible solutions can be inspected using the autoCONSTRAINT method in

order to provide mechanistic insight and to quantify the most likely rates of much of the dominant chemistry in the system. The larger the number of experiments and the broader the coverage of the chemical space, the greater the confidence in the rate coefficients determined by the approach.

As the method is, as yet, unable to solve the rate coefficients from several datasets simultaneously, they are determined one by one. To find a "good" set of rate coefficients, relative rate coefficients must be found that work for each dataset. To

395 overcome this problem, an algorithm to search the parameter space of the relative rate coefficients is in development. Where the types of chemical equations represent a good description of the dynamics in the system, at least one region in the parameter space of relative k-rates can be found reporting similar rate coefficients from all experiments. Note that these rate coefficients ideally are determined from dominant reaction terms (within equ. 2).

### Code & data availability

All input data to autoCONSTRAINT and PyCHAM, as well as all data shown in the results, are available online together with an autoCONSTRAINT code version to reproduce the data: https://doi.org/10.5281/zenodo.14223708
The latest version of the autoCONSTRAINT code can be obtained by contacting the corresponding author.

## Author contributions

LP constructed the autoCONSTRAINT method and developed the codes. SOM adapted the PyCHAM model. All authors were involved in designing the simulations, analysing the data and writing the manuscript.

## Competing interests

The authors declare that they have no conflict of interest.

## Acknowledgements & financial support

S. P. O'M. acknowledges ongoing support from the UK National Centre for Atmospheric Science. L. P., S. P. O'M. and G. M. acknowledge support from the UK National Environmental Research Council funded Secondary Organic Aerosol Prediction in Realistic Atmospheres (SOAPRA) project, grant number NE/V012665/1. L.P. acknowledges the support by Austrian Science fund (FWF) for working on autoxidation chemistry (grant no. J-4241 Schrödinger Programme).

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
