# Peer review of "Theory informed, experiment based, constraint on the rate of autoxidation chemistry – An analytical approach"

_Aerosol Research, 2024_

## Author Response (AR1)

We want to acknowledge the reviewer's time and effort to review the submitted manuscript. Further, we are thankful for the constructive criticism which was considered carefully to improve the mansucript. Find below the answers to the issues raised and changes made in the text. Changes in modified sentences are highlighted.

**Referee #1:**

Manuscript `ar-2024-40` (Pichelstorfer et al.) presents a new methodology to derive chemical mechanisms that describe auto-oxidation reactions. This class of reactions has gained attention recently because of their role in the oxidation of volatile organic compounds and in the formation of secondary organic aeerosol. Because of the complexity of the chemistry the representation of these reactions in chemical models is a difficult problem to solve. This study proposes a possible solution to the problem and the results are very promising. The manuscript is well written and the conclusions are well supported. Therefore I recommend publication after the authors have addressed the following points.

**MAIN POINTS**

**Comment #1:**

"The methodology described in this manuscript is complex and I fully appreciate that it is difficult to explain, but I think the paper could benefit from a bit more high-level explanation, in simple terms."

**Answer:**

"We agree and added a high-level explanation to the last paragraph of the introduction."

**Changes made in the manuscript:**

Addition to the last paragraph of the introduction: "The method compares change rates and concentrations of detected species (at atomic resolution, e.g., by NO3- CIMS) to isomers that exist in a proposed, lumped reaction scheme (Pichelstorfer et al., 2024). Assumptions on the relative magnitude of the isomer-forming rate coefficients and a description of physical change parameters (e.g., dilution, wall interaction) allow to calculate reaction rate coefficients of the proposed scheme."

**Comment #2:**

"As I understand it, for a given chemical system the model chooses a set of rate coefficients for the auto-oxidation reactions. Using this set, a certain distribution of reaction products is obtained that can be compared to the distribution of masses observed by the nitrate-CIMS. This may be too simplistic of an explanation but is it more or less correct?"

**Answer:**

"Yes, the steps described are correct. Creating a reaction scheme describing autoxidation chemistry, however, is not part of the current work but is described in more detail in Pichelstorfer et al., 2024. In this work, we focus on the question: how can we quantify the rate coefficients of the autoxidation scheme?

Yes, we agree that reaction products of the autoxidation scheme are then compared to masses observed by the nitrate-CIMS."

**Changes made in the manuscript:**

We added a high-level description to the end of introduction (see answer to **comment #1**)

**Comment #3:**

"How is the best fit to the experimental data determined?"

**Answer:**

"In the submitted manuscript, we do not investigate experimental data but simulated CIMS-data. To derive the results shown in Figs. 3 and Figs. S3-S6, we applied known relative quantities of the rate coefficients (for details see section *2.3 Solving the equations*). However, for CIMS data determined in an experiment, these quantities are unknown and, to date, can only be determined empirically. A potential approach is outlined in section *3.3 On determining multi-experiment based, realistic rate coefficients.*"

**Changes made in the manuscript:**

To stress the use of simulated data we changed in the introduction from "simulated mass-spectral data" to "simulated data".

Similarly, in the last section of the introduction: "simulated mass spectral observations" is replaced by "simulated mass-spectra".

**Comment #4:**

"The assumption of the method, if I understand it correctly, is that each mass measured by the nitrate-CIMS corresponds to a product of reactions R1-R6."

**Answer:**

"Whilst this unambiguity would be ideal, this can only be considered as the general case. There are some exceptions that must be considered: a) some detected masses don't correspond to a product of reactions R1-R6. These peaks are reported in the output files (file "getrate\_out.txt", see SI section autoCONSTRAINT output for more information) and may indicate reaction pathways not yet considered. b) several products of the scheme may relate to a single detected peak. See e.g., SI section Exemplary solution of equ. (2) for CIMS peak that has 4 model isomers.

Both a) and b) are already detailed in the manuscript text (second paragraph of section 2.3 *Solving the equations; and equ. 2 and description*)."

**Changes made in the manuscript:**

no changes made to the manuscript

**Comment #5:**

"My first question is: are you trying to keep a mass balance? In other words is the composition of these products in the model tracked, so that their mass can be calculated and matched to a measured mass?"

**Answer:**

"Yes, we are trying to keep the mass balance and track the composition of any species formed. Regarding mass balance, the only exceptions are reaction equations R2, where the sum of peroxy radicals is not changed although a H-atom is exchanged (a respective sentence can be found just below the list of reactions R2:  $RO_2 + \Sigma RO_2$ •)."

**Changes made in the manuscript:**

no changes made to the manuscript

**Comment #6:**

"If this is the case is it a problem that you are coupling this methodology with the MCM, a mechanism which doesn't not keep mass balance?"

**Answer:**

"The present approach determines rate coefficients **starting at the upper end of the mass spectrum** (see SI section "On sorting the input CIMS data" for details). The loss of the heaviest molecules is balanced by production terms (autoxidation reactions of RO or RO2; see equ. (3) for details). Accordingly, mass balance is a central aspect of the proposed method and it does not directly relate to MCM due to the order of the determination of the rate coefficients. MCM species are typically found at lower molecular masses and, by autoxidation reactions, provide influx to the APRAM scheme."

**Changes made in the manuscript:**

no changes made to the manuscript

**Comment #7:**

"How do you deal with different molecules that have the same mass? Mass spectrometric methods are often unable to distinguish between those and that may introduce a substantial bias in the methodology."

**Answer:**

"In the present approach, we assume that we can't distinguish between isomers in the experimental data. All isomers of the proposed scheme are added up when compared to a CIMS-peak (see equ. 2 and the related description and section "Exemplary solution of equ. (2)" in the SI). Accordingly, we account for this issue. If in future, information of isomeric distribution may be available, it will remove much of the ambiguity in the calculation of rate coefficients using the autoCONSTRAINT method."

**Changes made in the manuscript:**

no changes made to the manuscript

**Comment #8:**

" Likewise, fragmentation of molecules in the mass spectrometer may cause masses to be incorrectly assigned to a molecule, and that will also bias the results. Can the authors comment on these points?"

**Answer:**

"We do agree that detected fragmentation products may incorrectly be assigned to a molecule, and, as a consequence, rates may be biased. Since structures of the fragmenting molecules are unknown, the authors think there is no way to predict fragmentation products reliably. A respective statement can be found in the methods section (starting line 156).

Additional information on the treatment of fragmentation products in the autoCONSTRAINT method can be found in section *2.1 Key aspects of the approach* (see subsection *Fragmentation products*)."

**Changes made in the manuscript:**

We modified a sentence in the *Methods* section to clarify the treatment of fragmentation species (from line 154): "Concentrations of fragmentation products, due to the unknown fragmentation pathways, are estimated (lower limit is 0; upper limit is determined by the kinetic limitation for the considered reaction, e.g. RO2+HO2; for R6c, the upper limit is the production of RO species)."

**Comment #8:**

"Fragmentation of the reaction products is mentioned on page 7, and I agree with the authors that it has a potentially large impact on the final results. It would be good to add a comment on how much of a bias these processes may introduce in the methodology."

**Answer:**

"We quantify the impact of fragmentation products in the sentence starting from line 206 (For mass balance, the larger the estimated flux through fragmentation (see reaction equations 2d, 3c, 4d, 5c and 6c), the higher the production of the reagents via reaction 5a and 6a must be.). However, we too think some additional clarification may be helpful and added a section on the impact of fragmentation to the SI."

**Changes made in the manuscript:**

We added a section to the SI, p.7 ("Effect of the fragmentation pathway on the rate coefficients").

**Comment #9:**

"Are the fragmentation channels simply ignored? This may be acceptable as an approximation in a method that is still being developed, but needs to be clarified."

**Answer:**

"No, fragmentation channels are not ignored. See previous comments."

**Changes made in the manuscript:**

No changes made in the manuscript

**Comment #10:**

"Another concern I have is that the methodology described here relies on the assumption that the RO2 sum calculated by the MCM is correct, not to mention the distribution of the individual RO2. This is far from a given, and in fact I would argue is a major uncertainty of the MCM, especially for some compounds. Can you comment on this point?"

**Answer:**

"We agree that the presented application of autoCONSTRAINT method to some extent relies on the sum of RO2 (which consists of the sum of RO2 calculated using MCM as well as the sum of detected species covered by the autoxidation scheme) via reactions R1 and R2. As a consequence, the respective rate coefficients may be biased by errors in the RO2 sum and distribution of MCM RO2. This, however, does not impact the rate coefficients of all other reaction pathways due to the empirical nature of the approach: while the rate coefficients in R1 and R2 may be off, due to a wrong RO2 sum, the molecular flux through the respective channel is determined empirically (i.e., the change observed in a mass peak  $\rightarrow$  see first term of equ. 2). Note that is the molecular flux which is considered in the mass balance equ. 3 determining the H-shift rates of RO2 (R5a) and RO (R6a). Other rate coefficients (R3 – R6, except for R5a and R6a) are not affected as they are determined independently from MCM RO2 species.

When applying the derived autoxidation scheme in a different chemical regime, the error propagates to the calculated result for R1 and R2.

We, the authors, are aware of the fact that MCM-derived RO2 concentrations feature errors that impact the autoCONSTRAINT derived reaction schemes. As outlined in the introduction, yet, there is no method available to quantify autoxidation chemistry, except for a few reaction pathways. Accordingly, we think that the present work, although linked to a model like MCM, represents a step forward in the field of autoxidation and related (e.g., aerosol precursor formation)."

**Changes made in the manuscript:**

No changes made in the manuscript

**MINOR POINTS**

**Comment #11:**

"Lines 75-80: is there way to interpret the ion counts other than concentrations?"

**Answer:**

"Yes, actually, the interpretation of ion counts is a complex task, depending on the specificity and sensitivity of the chemical ionisation scheme and instrument configuration. The ion count represents the detected ions in the mass spectrometer and depends on several processes that may vary between different molecule structures, reagent ions as well as setups (see e.g., Hyttinen et al., 2018) "

Hyttinen, N., Otkjær, R. V., Iyer, S., Kjaergaard, H. G., Rissanen, M. P., Wennberg, P. O., and Kurtén, T.: Computational Comparison of Different Reagent Ions in the Chemical Ionization of Oxidized Multifunctional Compounds, J. Phys. Chem. A, 122, 269–279, https://doi.org/10.1021/acs.jpca.7b10015, 2018.

**Changes made in the manuscript:**

no changes made in the manuscript

**Comment #12:**

"Lines 89-91: check punctuation and open/closed brackets."

**Answer:**

"Thanks, for highlighting the issue."

**Changes made in the manuscript:**

We changed the respective sentence (changes highlighted): "Generally, each peroxy radical can interact with any other peroxy radical to either form accretion products (ROOR': Berndt et al., 2018b), alkoxy radicals (Crounse et al., 2013), or form hydroxyl (ROH) and carbonyl (RC=O) species in a complex chemical reaction (see: Orlando and Tyndall, 2012; Salo et al., 2022)"

**Comment #13:**

"Line 299: correct "datat"."

**Answer:**

"Thanks, for highlighting the misspelling"

**Changes made in the manuscript:**

We changed to "data".

**Comment #14:**

"Supplement: ensure MCM is capitalized."

**Answer:**

"Thanks, for highlighting, we will took care that MCM is capitalized throughout the Supplement."

**Referee #2:**

**General comments**

Overall, this is a very well written and clearly structured manuscript about a novel semi-automated method used to constrain autoxidation reaction schemes, including both unimolecular peroxy/alkoxy radical reactions and all related biomolecular reactions that influence the autoxidation reaction mechanisms. The presented novel autoCONSTRAINT tool seem to be a potentially very useful software for the construction of realistic, theoretically consistent and clearly documented and reproduceable peroxy/alkoxy radical autoxidation mechanisms for a large number of VOC + oxidation systems. Currently such a tool is not existing and hence this manuscript provide a substantial contribution to scientific progress within the scope of atmospheric chemistry and secondary organic aerosol formation.

**Specific comments**

**Comment #1:**

"L105 "Reactions R2a to R2c do only change the reacting peroxy radical and have no effect on the  $\Sigma$ RO2•." Consider to reformulate this sentence. Reactions R2a to R2c only influence the concentration of the reacting peroxy radical and has only minor effect on the  $\Sigma$ RO2•

I think that reacting peroxy radical is also part of  $\Sigma$ RO2, so some minor impact R2a to R2c should also have on the  $\Sigma$ RO2•or am I wrong?"

**Answer:**

We fully agree. Reactions R2a – R2c do effect the  $\Sigma$ RO2 by reducing the reacting peroxy radical's concentration. We will reformulate the respective sentence as suggested.

**Changes in the manuscript:**

The respective sentence is reformulated (p.4, l. 108): "Reactions R2a to R2d do change the reacting peroxy radical's concentration and have only minor effect on the  $\Sigma RO_2$ •."

**Comment #2:**

"Page 4: Why do you consider that fragmentation products can form from RO2• + HO2•and RO2• + NO•reactions but not RO2• +  $\Sigma$ RO2•?"

**Answer:**

Thanks for pointing out this inconsistency. The reaction pathway RO2• +  $\Sigma$ RO2• leading to fragmentation of the product was omitted unintentionally in the text and, actually, is part of the reaction pathways covered by the current version of the autoAPRAM-fw. We will add the missing reaction as reaction R2d to the manuscript.

**Changes in the manuscript:**

The following additions/changes are made, starting from reaction equation (R2c):

(R2d)

Reactions R2a to R2d do change ..."

**Comment #3:**

"L260: "In the current work we assumed  $k_{autox,high} = 2 \text{ s}^{-1}$ . We chose this conservative upper limit as most H-shifts are significantly slower". What would the consequences be if you would allow considerably higher autoxidation reaction rates? "

**Answer:**

The consequence would be a reduction in the expected initiation flux (i.e., production of the species  $RO_{2,k}$  from MCM species). As outlined in the last paragraph of section 2.2, the "…H-shift of RO• and RO2•, together with the initiation-flux form an equation with three unknowns." Accordingly, a single experimental data-set will only allow for determining the range of production of  $RO_{2,k}$  but not for determining specific magnitudes of the fluxes. The higher the considered  $k_{autox,high}$ , the larger will be the 2D surface of potential solutions. However, equ. (4) & (5) do not aim to quantify the influx required but aim to highlight situations where influx very likely is needed.

**Comment #4:**

"Page 9: The description and use of the potential initiation flux (P), equ. 4-5 need to be described more clearly. It is not easy to understand how equ. 4-5 was derived and what these equations represents. I guess that they are derived from equ. 3 or? These equations do not seem to represent the potential initiation flux (units of molecules cm-3 s-1) but ratios (potentials) of how much RO autoxidation and RO2 autoxidation may contribute to the production of a specific RO2 peak."

**Answer:**

We agree. First of all, we will change the name from "potential initiation flux", which indicates a mass flux to "initiation flux probability". Further we will add a description of how this probability relates to equ. (3) to relate the potential to prior considerations.

**Changes in the manuscript:**

In the sentence above equ. (4): "The initiation flux probability (P) ..."

In equ. (4), there was a mistake (missing "1 - …), thus, we added "1 - …" : "P(RO•) = 1 - ( $L_{chem} + L_{phys} - S_{in}$ ) / ( $L_{chem} + L_{phys} + S_{in}$ )"

From line 263 we add: "Sin is a source term of the RO2,k• via the alkoxy pathway."

From line 267 we add: "The initiation flux probabilities of RO• and RO2• relate to equ. (3) as all are based on considerations of mass conservation. While equ. (3) is explicitly balancing mass fluxes, P is meant to highlight the imbalances in fluxes by considering a) mass fluxes via the RO pathway for P(RO•), and b) expected limitations of the fluxes from  $RO_2•$  within the autoAPRAM scheme for  $P(RO_2•)$ ."

**Comment #5:**

"Should not equ. 3 also have a separate term for the RO autoxidation source of  $RO_{2,k}$  or is the RO autoxidation source included in  $S_k$ ?"

**Answer:**

Yes, the equ. (3) has to consider the source of  $RO_{2,k}$  via the alkoxy pathway and, yes, this pathway is part of the term  $S_k$ . We will add a statement specifying explicitly that RO autoxidation is part of  $S_k$  to increase clarity.

**Changes in the manuscript:**

Sentence starting from line 217 is reformulated: "Where C(RO2,k•) is the concentration of the peroxy radical k. Sk is a source term considering all potential contributions including wall sources as well as autoAPRAM-RO• autoxidation, MCM-RO• and MCM-RO2• pathways. Lk is a loss term (physical as well as chemical but not including RO2• autoxidation)."

**Comment #6:**

"I guess you don't need to constrain the RO autoxidation rate since this is assumed to be very fast. Is this why it is omitted in equ. 3-5?"

**Answer:**

Yes, we assume that the alkoxy reactions are very fast (~  $10^6$  s-1). As a result, the limiting rate is the formation of alkoxy radicals. This seems to be true at least for the model data, where reactions of the alkoxy radicals are considered explicitly (i.e., they have a specific rate) while in the analysis, using autoCONSTRAINT, we consider instantaneous formation of the products via the alkoxy pathway. The results (Fig. 3 and Figs. S3 to S6) show that this assumption does not introduce a notable error (the maximum error found among all rate coefficients determined is roughly 0.5% which will not matter in any experimentally derived results where much larger errors are expected to result from the measurement).

**Technical corrections**

**Comment #7:**

L34: "a drop of the saturation vapor pressure". Consider if it maybe better to write a decrease in the saturation vapor pressure.

**Answer:**

We changed from "drop" to "decrease"

**Comment #8:**

L106-107: ... those equations difficult to constrain. Change to: those equations difficult to constrain that are difficult to constrain.

**Answer:**

We changed from "... those reaction equations difficult to constrain." to "... those reaction equations that are difficult to constrain."

**Comment #9:**

L203: "Although direct inclusion of the fragmentation products is beyond the scope of this work, the effect of this reaction can be investigated." Change to:

"Although direct inclusion of reactions leading to fragmentation products is beyond the scope of this work, the effect of such reactions can be investigated."

**Answer:**

The reactions leading to the formation of fragmentation products are included. However, predicting the reactant-specific atomic composition of fragmentation products is beyond the scope of this work.

**Changes in the manuscript:**

We reformulate the respective sentence: "... inclusion of reactant-specific fragmentation products is beyond the scope of this work ..."

**Comment #10:**

L272: "see Fig. Worklfow\_1.pdf". Change to see Fig. 2

**Answer:**

we changed to "Fig. 2"

**Comment #11:**

L272-273: "The autoCONSTRAINT tool reads in a chemical scheme with no rate coefficients, together with a NO3- obtained mass spectrum" I am not exactly sure what you want to state with this sentence, but would it not be better to write:

The autoCONSTRAINT tool reads in a chemical scheme with no rate coefficients, together with a mass spectrum obtained e.g. with a NO3- CIMS.

**Answer:**

we agree and changed accordingly

**Changes in the manuscript:**

Line 280: "The autoCONSTRAINT tool reads in a chemical scheme with no rate coefficients, together with a mass spectrum obtained e.g. with a  $NO_3^-$  CIMS."

**Comment #12:**

L299: "datat" should be data

**Changes in the manuscript:**

"datat" was changed to "data"